# Optical quality of Fresnel-structured hyperopic implantable collamer lenses: A ray-tracing analysis

**Kimiya Shimizu**[1,2☯]**, Takushi Kawamorita**[3*☯]

**1** Department of Eye Center, Sanno Hospital, Minato-ku, Tokyo, Japan, **2** International University of Health and Welfare, Minato-ku, Tokyo, Japan, **3** Department of Orthoptics and Visual Science, Kitasato University, Sagamihara, Kanagawa, Japan

☯ These authors contributed equally to this work.
* kawa2008@kitasato-u.ac.jp

**Data availability statement:** All relevant data are within the manuscript.

**Funding:** This study was supported solely by institutional research funding from Kitasato University (1070111602).

## Abstract

To achieve positive refractive power, implantable collamer lenses (ICLs) for hyperopia increase central thickness, which in turn raises the risk of pupil blockage and lens contact in eyes with shallow anterior chambers. Additionally, incorporating a central hole raises concerns about stray light and halo formation due to increased thickness. This study used optical simulation to evaluate the imaging performance of hyperopic ICLs, highlighting the Fresnel structure's potential to reduce thickness while maintaining imaging performance. Optical simulations were performed using Ansys Zemax OpticStudio 2024 R2 (Ansys, Inc.). An ICL with +10.00 D correction was designed for the Liou & Brennan model eye. The vault was set to 0.50 mm, and the central hole diameter was set to 0.36 mm. Fresnel ICLs (Fr. ICLs) with step heights ranging from 25 to 100 μm were compared with the reference ICL (Ref. ICL). Modulation transfer function (MTF) analyses were performed at internal artificial pupil diameters of 3.00 mm and 4.50 mm, both with and without holes. By reducing the central thickness from 0.80 mm to 0.40 mm, the Fr. ICL for hyperopia showed higher MTF values than the Ref. ICL at both pupil diameters. The difference was particularly significant at 4.50 mm. A slight MTF decrease due to the central hole was observed at 3.00 mm internal artificial pupil diameter. A step height of 50 μm or more suppressed contrast reduction at high spatial frequencies. Although further research is needed in illumination optics and clinical studies, ray-tracing simulations have demonstrated that the designed hyperopia-correcting Fr. ICL can reduce central thickness while maintaining or improving optical performance, making it a promising candidate for clinical application.

**Competing interests:** Kimiya Shimizu reports grants, personal fees, and non-financial support from STAAR Surgical Company; personal fees and non-financial support from Kowa Company, Ltd.; grants, personal fees, and non-financial support from Santen Pharmaceutical Co., Ltd.; grants and non-financial support from Senju Pharmaceutical; and personal fees and non-financial support from HOYA Corporation, outside the submitted work. Takushi Kawamorita reports grants and non-financial support from Denso Corp.; grants and non-financial support from Nissan Motor Co., Ltd.; grants and non-financial support from Tbwa\Hakuhodo; grants and personal fees from Wakamoto Pharmaceutical Co., Ltd.; grants and personal fees from Johnson & Johnson; grants and personal fees from CooperVision, Inc.; grants and personal fees from Menicon Co., Ltd.; grants and personal fees from Itoh Optical Industrial Co., Ltd.; personal fees from Alcon Inc.; personal fees from STAAR Surgical Company; personal fees from Kowa Company, Ltd.; personal fees from Santen Pharmaceutical Co., Ltd.; personal fees from Senju Pharmaceutical; personal fees from Bausch & Lomb Incorporated; personal fees from ROHTO Pharmaceutical Co., Ltd.; personal fees and non-financial support from HOYA Corporation; and personal fees from Beaver-Visitec International, Inc., outside the submitted work. This does not alter the authors' adherence to PLOS ONE policies on sharing data and materials.

## Introduction

Implantable collamer lenses (ICLs) are widely used as a refractive correction method, enabling vision restoration even in cases unsuitable for corneal refractive surgery. In particular, posterior chamber ICLs have demonstrated favorable clinical outcomes for moderate to severe myopia, including long-term follow-up [1–3].

Conversely, ICLs for hyperopia have demonstrated good optical properties [4,5] and clinically favorable outcomes [6]. However, to achieve positive refractive power, the central thickness of the lens increases, which may result in pupil blockage [7] or increased risk of contact with the crystalline lens in cases where the anterior chamber depth is insufficient. This results in limitations regarding vault maintenance, surgical maneuverability, and management of complication risks, thereby posing barriers to clinical adoption. Additionally, increased central thickness may raise concerns about increased light scattering and halo formation, especially if a hole is created in the central region, as this enlarges the cross-sectional area of the hole.

However, hyperopia exists across a wide age range, from children to adults and the elderly, and particularly in children, numerous cases of amblyopia caused by high hyperopia or anisometropic hyperopia have been reported [8]. In these cases, correction with glasses or contact lenses may not improve visual function, and in particular, for treatment-resistant anisometropic amblyopia, surgical options that provide permanent and stable refractive correction are required. ICL has been reported to be a safe and effective option for the treatment of anisometropic amblyopia [9,10]. Additionally, hyperopia is associated with computer vision syndrome [11]. In middle-aged and older adults, where accommodative ability decreases with age, hyperopia demands greater accommodation for near vision, potentially affecting quality of life and increasing the need for refractive correction.

Against this backdrop, efforts are underway to improve safety and expand the indications for ICLs through design modifications. One example is the Hole ICL developed by Shimizu et al., which features a small perforation in the central region [12]. This design eliminates the need for iridotomy [13] while maintaining aqueous humor circulation [14] and has been clinically applied. This structure exemplifies a design innovation that reduces the risk of intraocular pressure elevation and minimizes postoperative burden on patients, and may also contribute to reducing the risk of secondary cataract formation. However, this hole may cause stray light, potentially resulting in glare and halos. In particular, myopic ICLs have a thinner central optical zone, whereas hyperopic ICLs have a thicker one. Consequently, the greater thickness of hyperopic hole ICLs may lead to increased stray light generated at the hole.

In this study, we addressed the thickness issue of hyperopic ICLs by introducing a stepped-surface design inspired by refractive Fresnel optics. In the present study, the term "Fresnel ICL (Fr. ICL)" refers to a stepped refractive approximation of a continuous monofocal surface for thickness reduction and does not imply a diffractive Fresnel lens designed for interference control. A refractive Fresnel lens approximates a continuous refractive surface by segmenting it into concentric annular steps, which allows a substantial reduction in lens thickness while preserving the intended refractive optical power under the design condition. Such stepped-surface concepts

have been widely employed in illumination optics and, under certain conditions, have also been applied to imaging optical systems.

In contrast, this stepped refractive approach should be clearly distinguished from diffractive optical elements (DOEs). Diffractive multifocal intraocular lenses and extended depth-of-focus (EDOF) intraocular lenses [15], as well as presbyopia-correcting Extended-Depth-of-Focus ICL [16], may exhibit zone-based structures that appear superficially similar; however, their optical behavior is governed by interference effects and phase distributions [17]. Historically, the concept of "Fresnel zones" is rooted in phase repetition and interference; however, the Fresnel ICL proposed here adopts only the stepped-surface approximation for thickness reduction, without employing phase-based diffractive control. The hyperopic stepped ICL proposed in this study is not intended to increase the number of focal points or to extend depth of focus, but rather to reduce lens thickness while maintaining a single focal point.

In this study, we designed a Fresnel structure for a hyperopic ICL to reduce the thickness of the optical center to a level comparable to that of the lens periphery. To investigate the effects of this structure on visual performance, we quantitatively evaluated the modulation transfer function (MTF) using optical simulations. This study reports on the optical characteristics of the Fresnel structure-based ICL for hyperopia, discusses its design advantages and limitations, and explores its potential applications. We also assessed the sensitivity of optical performance to representative misalignment parameters to reflect clinically relevant conditions.

## Methods

Optical simulations were performed using Ansys Zemax OpticStudio 2024 R2 (Ansys, Inc.) in sequential mode. The base eye model was based on the Liou & Brennan model eye, which incorporates biometric measurements, accounts for asphericity in the cornea, and reflects the refractive index distribution of the crystalline lens [17]. The designed ICL was modeled to closely resemble the existing hole ICL KS-AquaPORT™ (STAAR Surgical, Nidau, Switzerland). The refractive error was set to +10.00 D at the spectacle plane, and a hyperopic ICL was designed to correct this error. The ICL position was set 0.50 mm toward the corneal side from the anterior lens position, with this configuration referred to as vaulting. The α angle and pupil eccentricity were not included in the baseline condition, and evaluations were conducted along the optical axis; their effects were separately assessed in the sensitivity analysis. Additionally, the presence or absence of a central hole was verified. The central hole has a diameter of 0.36 mm (radius 0.18 mm); therefore, in the optical design software, the central light beam was blocked by an aperture for analysis, following the approach described in previous reports [18] (Table 1).

As part of the design procedure, an internal artificial pupil (aperture stop) at the iris plane was set to 3.0 mm, and optical performance was additionally evaluated at 4.5 mm, in accordance with ISO 11979−2 and/or previously reported optical evaluation methods [20,21]. The object position corresponding to the far point was set to +10.00 D with a 12.00-mm vertex distance from the spectacle plane. The vitreous length was optimized using the root mean square (RMS) wavefront, and a refractive error of +10.00 D was created. Subsequently, a Ref. ICL was implanted, the object position was changed to infinity, and the ICL surface was optimized (Fig 1). The thickness of the Ref. ICL was set to 0.80 mm. The total axial length of the eye was 20.364 mm.

The Fr. ICL was designed based on the Ref. ICL. First, a relief with uniform step heights was added to the front of the ICL as a user-defined surface on top of the standard surface corresponding to the substrate. A rotationally symmetric stepped relief was applied, and the shape of the relief was calculated using even-order non-spherical polynomials. The sag of the surface is expressed by Equation 1.

$$z = z_{sub} + z_{relief} = \frac{cr^2}{1 + \sqrt{1 - (1 + k)c^2 r^2}} + mod\left(a_1 r^2 + a_2 r^4 + a_3 r^6 + a_4 r^8 + a_5 r^{10}, h\right)$$

(1)

Table 1. Improved Liou & Brennan model eye including ICL [19].

| Medium | Refractive index | Curvature radius | Conic coefficient | Thickness | Abbe number |
|---|---|---|---|---|---|
| Air | 1.000 | ∞ | 0.00 | N/A | Undefined |
| Cornea | 1.376 | 7.77 | −0.18 | 0.50 | 56.28 |
| Aqueous humor | 1.336 | 6.40 | −0.60 | 1.80 | 52.72 |
| ICL | 1.450 | 4.38 | 0.00 | 0.80 | 37.00/50.00* |
| Vaulting | 1.336 | 8.25 | 0.00 | 0.50 | 52.72 |
| Lens cortex | Grad A | 12.40 | 0.00 | 1.59 | GRIN** |
| Lens nucleus | Grad P | ∞ | 0.00 | 2.43 | GRIN** |
| Vitreous body | 1.336 | −8.10 | 0.00 | 12.74 | 53.33 |
| Retina | 1.336 | −12.00 | 0.00 | 0.00 | 50.20 |
| Item | Grad A*** | | Grad P*** | | |
| $N_{0,0}$ | 1.36800 | | 1.40700 | | |
| $N_{0,1}$ | 0.04906 | | 0.00000 | | |
| $N_{0,2}$ | −0.01543 | | −0.00661 | | |
| $N_{1,0}$ | −0.00198 | | −0.00198 | | |

* ICL material dispersion was evaluated using two Abbe numbers (37 and 50).

** The lens cortex and nucleus were modeled as gradient-index (GRIN) media; therefore, a single Abbe number was not defined.
*** Grad A and Grad P denote the anterior and posterior GRIN distributions of the crystalline lens, respectively.

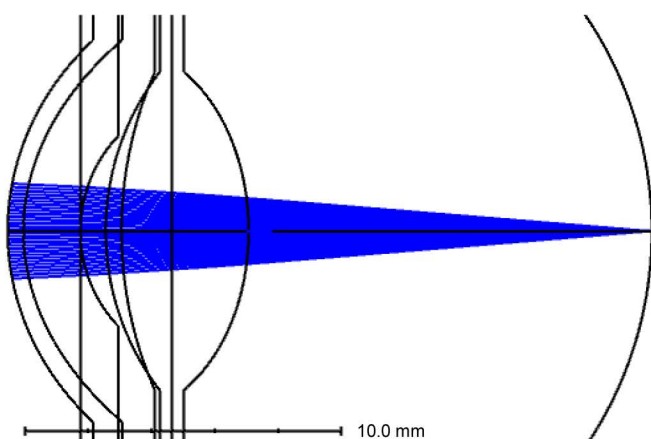
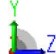

Fig 1. Layout of the improved Liou & Brennan model eye [19] with an implanted ICL in Ansys Zemax (Ansys Inc.).

Here, $z_{sub}$ denotes the base substrate sag, $z_{relief}$ the stepped relief term, mod the remainder function, c the curvature (reciprocal of the radius of curvature), k the conic constant, r the radial coordinate, and h the uniform step height of the relief.

In this study, the step height was set in 25 μm increments from 25 μm to 100 μm, and the front surface of the ICL was optimized by optimizing only the quadratic terms of the above equation. This step-height range (25–100 μm) was not intended to represent an optimized final clinical design; rather, it was selected to span a manufacturable and biomechanically plausible design space for Fr. ICLs, enabling a parametric evaluation of the influence of step height on optical performance and robustness. The ICL shape was optimized for an internal artificial pupil diameter of 3.0 mm. No further optimization was performed when the pupil diameter was changed to 4.5 mm, and comparisons were made under each pupil condition.

To reflect the Stiles-Crawford effect, the Gaussian apodization factor α was set to 0.054 at the pupil plane (Equation 2) [22].

$$zT(\rho) = e^{-\alpha^2}$$

(2)

Here, T denotes the transmission as a function of the position of the pupil and ρ the radial pupil coordinate.

Using both lenses, the modulation transfer function (MTF) at the image plane was analyzed using an FFT-based wave-optical method with a sampling resolution of 1024 × 1024. The analysis was performed under both monochromatic conditions (555 nm) and polychromatic (white-light) conditions to better reflect clinically relevant viewing environments. Because this study is based on a rotationally symmetric model, the reported MTF represents the average of the sagittal and tangential planes.

In addition to the modulation transfer function (MTF), the point spread function (PSF) and the Strehl ratio were evaluated as complementary indicators of image spread related to stray light, halo, and glare tendencies. All optical metrics were calculated using FFT-based wave-optical analysis to ensure methodological consistency across all conditions.

Polychromatic optical performance was assessed by performing FFT-based PSF and MTF calculations at five discrete wavelengths (470, 510, 555, 610, and 650 nm). These wavelengths were weighted according to the photopic luminous efficiency function, with relative weights of 0.091, 0.503, 1.000, 0.503, and 0.107, respectively. The Strehl ratio was derived from the FFT-based PSF.

To evaluate robustness against alignment errors and changes in optical conditions, contrast at a spatial frequency of 50 c/mm was calculated for the Ref. ICL and Fr. ICLs (step heights of 50 μm and 100 μm). Each parameter was varied independently, with other conditions fixed at reference values. The evaluated parameters were: axial displacement (ΔACD), ICL decentration, ICL tilt, pupil decentration, and alpha angle. The variation ranges for each parameter were set according to the conditions shown in Fig 8: ΔACD ranged from −0.4 to +0.4 mm, ICL decentration from 0 to 1.0 mm, ICL tilt from 0 to 10°, pupil decentration from 0 to 1.0 mm, and alpha angle from 0 to 10°.

## Results

The Fr. ICL was optimized based on an improved Liou & Brennan model eye that included ICL. Consequently, the front surface of the ICL became flatter, with the radius of curvature increasing to 5.23 mm, and the second-order term of the Fr. ICL surface was 0.025, indicating high imaging quality. Subsequently, while maintaining the vaulting, the thickness of the ICL was reduced to 0.40 mm. Thus, the total axial length remained at 20.364 mm.

The sag data excluding the base radius of the designed Ref. ICL and Fr. ICL are shown in Fig 2 and 3. It was confirmed that increasing the step height expands the range of the first zone and reduces the number of steps.

In addition, the first zone (radius) of the Fr. lens, as shown in Table 2, changed accordingly. It was found that as the step height increased, the radii of both the first zone and second zones also increased.

As shown in Fig 4, the MTF at 555 nm showed higher values for Fr. ICL than for Ref. ICL under both artificial pupil diameter conditions of 3.00 mm (Fig 4a) and 4.50 mm (Fig 4b). Among the Fr. ICLs, a tendency for MTF to decrease was observed under the 25 μm step height condition, and this difference was larger at an artificial pupil diameter of 4.50 mm (Fig 4b). Furthermore, the 50 μm step height condition showed slightly lower MTF values compared to the 100 μm condition. Regarding the effect of the hole, MTF with the hole was slightly lower than without the hole under most conditions, indicating that the degradation due to the hole was small and consistent.

Fig 5 showed the MTF of the Ref. ICL and Fr. ICLs with different step heights under multicolor conditions. At the 3.0 mm pupil condition, the Fr. ICL with a 25 μm step height exhibited lower MTF values than the other Fr. ICLs at most spatial frequencies, regardless of the Abbe number. Under the 4.5 mm pupil condition, the Fr. ICLs with step heights of 50 μm and 100 μm maintained higher MTF than the Ref. ICL across a broad range of spatial frequencies. This trend was consistently observed for both Abbe numbers of 37 and 50.

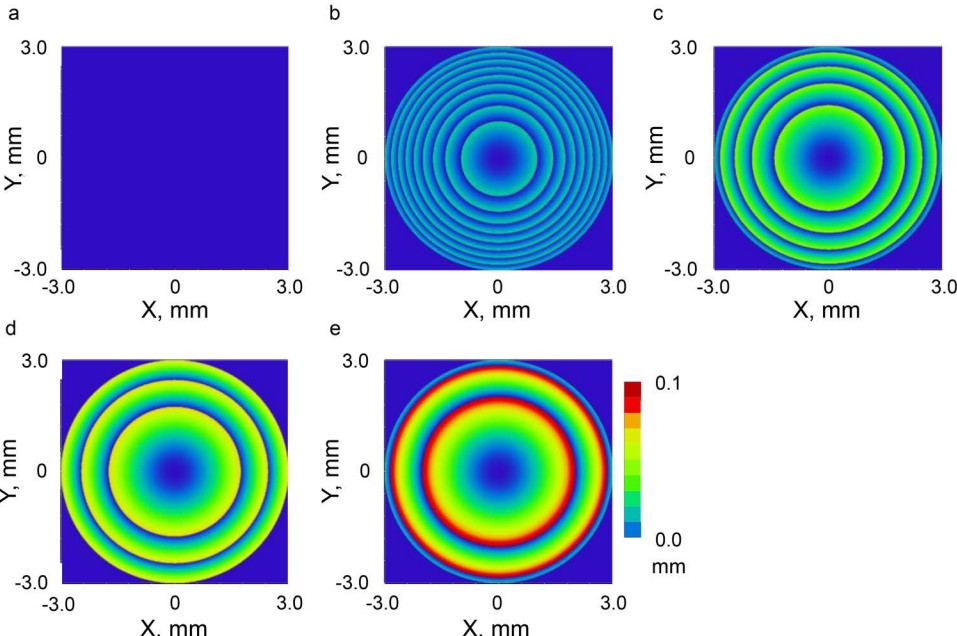

**Fig 2. Cross-sectional sag profiles of the Ref. ICL and Fr. ICLs (excluding the base radius). (a) Ref. ICL; (b-e) Fr. ICLs with step heights of 25, 50, 75, and 100 μm.**

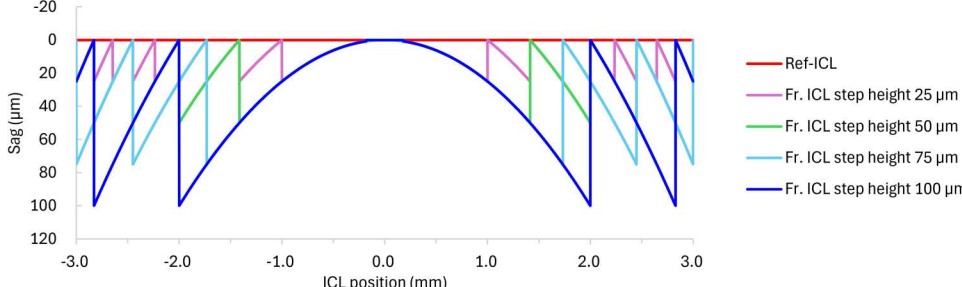

**Fig 3. Cross-sectional sag profiles of the Ref. ICL and Fr. ICLs (excluding base radius). Sag values are positive in the direction of light propagation, and the vertical axis is inverted for visualization.**

**Table 2. Radii of the first and second zones of the Fr. ICL as a function of step height.**

| Fr. ICL step height, μm | 1st zone position, mm | 2nd zone position, mm |
| --- | --- | --- |
| 25 | 1.00 | 1.41 |
| 50 | 1.41 | 2.00 |
| 75 | 1.73 | 2.45 |
| 100 | 2.00 | 2.83 |

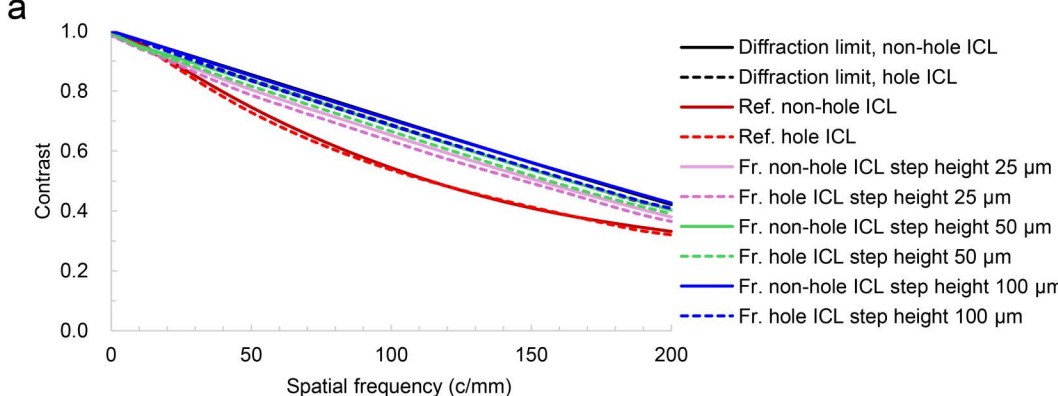

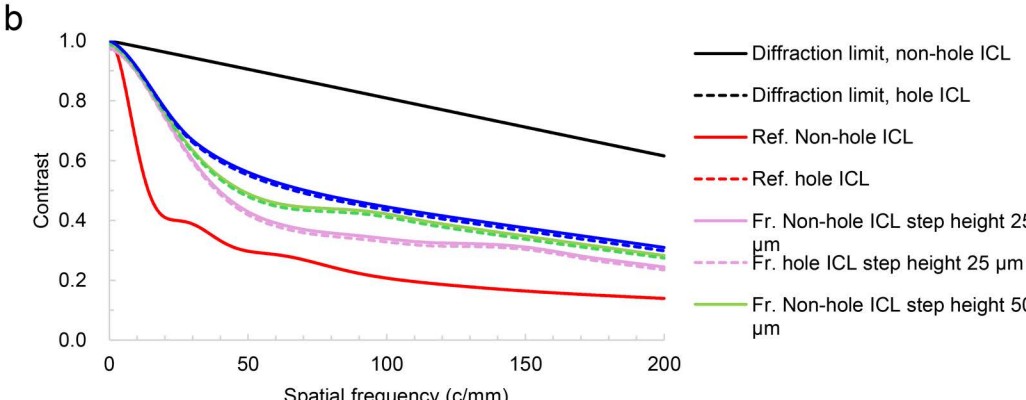

**Fig 4. MTF curves of the Ref. ICL and the Fr. ICLs at 555 nm for artificial pupil diameters of 3.00 mm (a) and 4.50 mm (b).** Solid and dashed lines indicate non-hole and hole designs, respectively.

Table 3 shows the FFT-derived Strehl ratios of the +10 D Ref. ICL and Fr. ICLs with different step heights at artificial pupil diameters of 3.0 mm and 4.5 mm. Under monochromatic light conditions at 555 nm, all Fr. ICLs showed higher Strehl ratios than the Ref. ICL at both pupil diameters, and the Strehl ratio increased with increasing step height. In contrast, under white light conditions, the Fr. ICL with a 25 μm step height showed a significant decrease in Strehl ratio compared to the Ref. ICL at a pupil diameter of 3.0 mm, whereas the Fr. ICLs with step heights of 50 μm and 100 μm maintained high Strehl ratios. At a pupil diameter of 4.5 mm, the Strehl ratio increased with increasing step height under both monochromatic and white light conditions.

Fig 6 shows the FFT-derived two-dimensional PSF at an artificial pupil diameter of 3.0 mm. Under monochromatic light conditions at 555 nm (Fig 6a), Fr. ICLs with step heights of 50 μm and 100 μm exhibited high central peak intensities and reached the maximum value on the normalized scale, whereas the Ref. ICL and the 25 μm step height showed relatively low central peak intensities. Under white light conditions (Fig 6b), this trend became more pronounced, with a clear decrease in the central peak observed for the 25 μm step height, which did not reach the maximum intensity. In contrast, the 50 μm and 100 μm step heights maintained high central peak intensities even under white light conditions.

Fig 7 shows the FFT-derived two-dimensional PSF at an artificial pupil diameter of 4.5 mm. Under monochromatic light conditions at 555 nm (Fig 7a), Fr. ICLs showed a tendency for central peak intensity to increase with increasing step height, with the 100 μm step height exhibiting the highest peak intensity. A similar trend was observed under white light

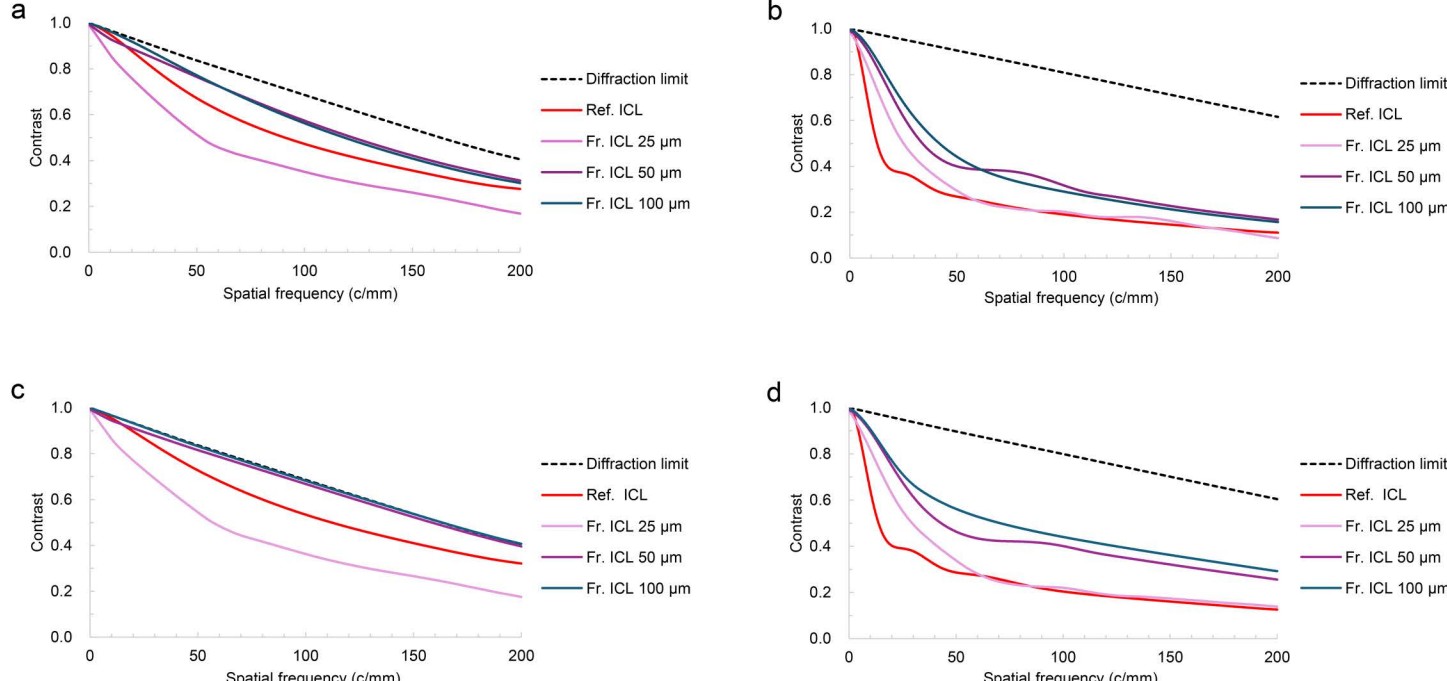

**Fig 5. Polychromatic MTF curves for Ref. ICL and Fr. ICLs with different step heights.** The dashed line indicates the diffraction limit. The Ref. ICL is shown in red, and Fr. ICLs with step heights of 25, 50, and 100 μm are shown in light pink, purple, and blue, respectively. **(a)** 3.0-mm pupil, Abbe 37; **(b)** 4.5-mm pupil, Abbe 37; **(c)** 3.0 mm pupil, Abbe 50; **(d)** 4.5 mm pupil, Abbe 50.

**Table 3. FFT-derived Strehl ratios for the +10 D Ref. ICL and Fr. ICLs with a central hole at artificial pupil diameters of 3.0 and 4.5 mm.**

| Artificial pupil diameter (mm) | Illumination (555 nm/ Polychromatic) | Ref. ICL | Fr. ICL (25 μm) | Fr. ICL (50 μm) | Fr. ICL (100 μm) |
|---|---|---|---|---|---|
| 3.0 | 555 nm | 0.808 | 0.845 | 0.938 | 0.990 |
| 4.5 | 555 nm | 0.136 | 0.259 | 0.332 | 0.377 |
| 3.0 | polychromatic | 0.799 | 0.455 | 0.918 | 0.980 |
| 4.5 | polychromatic | 0.127 | 0.194 | 0.315 | 0.385 |

conditions (Fig 7b), where the Reference ICL and 25 μm step height showed low central peak intensities, while the 50 μm and 100 μm step heights maintained high central peak intensities.

Fig 8 shows the results of comparing contrast at 50 c/mm against various parameter changes. When ΔACD was varied, the contrast showed a bell-shaped change, and across the entire range, Fr. ICLs (step heights of 50 μm and 100 μm) showed higher values than Ref. ICL, with 100 μm showing the highest tendency in particular. As the ICL decentration amount increased, the contrast decreased. The decrease in Ref. ICL was the greatest, whereas Fr. ICLs maintained relatively high contrast even under conditions of large decentration, with 100 μm showing the highest values. As ICL tilt increased, contrast decreased. In the low-tilt range, the Fr. ICLs, particularly the 100 μm design, showed higher contrast than the Ref. ICL. However, this difference became smaller with increasing tilt, and the contrast values converged in the high-tilt range (approximately 8–10°). With pupil decentration (0–1.0 mm), the contrast remained nearly constant, and Fr. ICLs (50 μm, 100 μm) maintained higher values than Ref. ICL. As the alpha angle increased, the contrast decreased; in the low to mid-angle range,

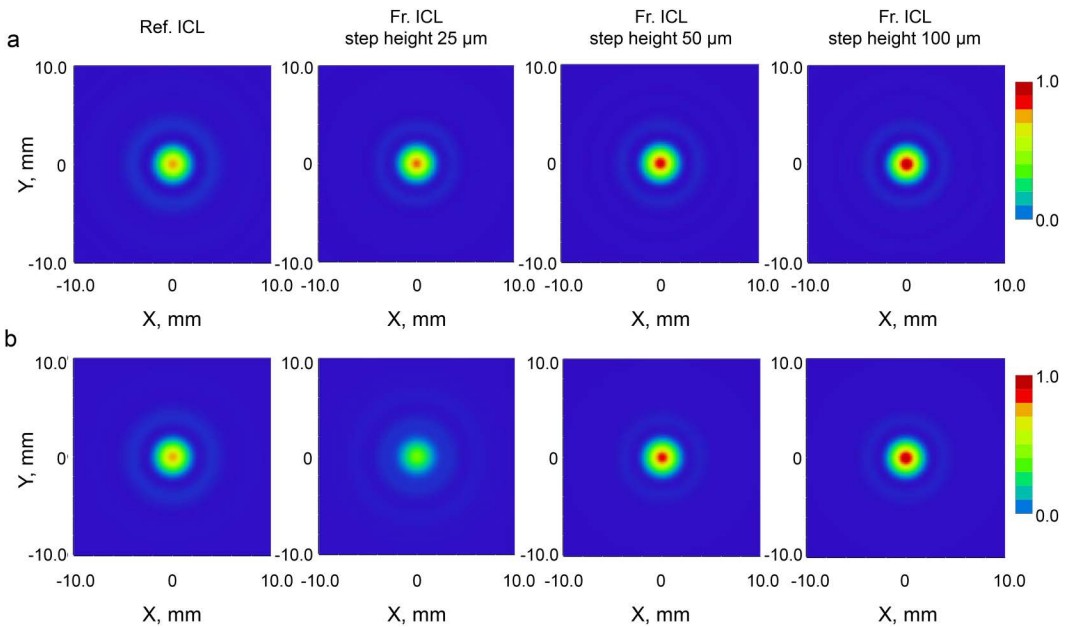

**Fig 6. Two-dimensional point spread functions (PSFs) derived from FFT calculations for the Ref. ICL and Fr. ICLs with different step heights.** (a) Monochromatic illumination at 555 nm with an artificial pupil diameter of 3.0 mm. (b) Polychromatic illumination with an artificial pupil diameter of 3.0 mm. In each row, images are shown from left to right: Ref. ICL, and Fr. ICLs with step heights of 25 µm, 50 µm, and 100 µm. The color scale ranges from 0 to 1 and represents the normalized peak intensity; lenses with lower Strehl ratios do not reach the maximum (red) intensity.

Fr. ICL 100 µm showed a tendency to be higher than Ref. ICL, while in the high-angle range, there were regions where Fr. ICL 50 µm fell below Ref. ICL, and Fr. ICL 100 µm generally maintained values equal to or higher than Ref. ICL.

## Discussion

In this study, MTF analysis of a hyperopic Fr. ICL was performed using optical simulation. Under idealized simulation conditions, reducing lens thickness increased the degrees of freedom for optimizing the anterior surface shape, resulting in slightly higher MTF values compared to the Ref. ICL. The effect of the central hole on MTF was small, which is consistent with previous reports, confirming that image quality can be maintained even with a hole in the hyperopic Fr. ICL. Furthermore, these trends were consistently observed under both monochromatic (555 nm) and polychromatic (white light) conditions. Additionally, within the evaluated range, the hyperopic Fr. ICL was suggested to be relatively robust.

In MTF analysis, the higher MTF values observed in the Fr. ICL compared with the Ref. ICL are attributed to the reduction in thickness and optimization of the front surface through flattening, which likely improved wavefront quality, as supported by the Strehl ratio results in Table 3. This difference became more pronounced when the pupil size increased, as aberrations tend to increase under such conditions. In addition, a decrease in contrast sensitivity has been reported in diffractive multifocal intraocular lenses, which have a similar relief type [17,23]. This is a phenomenon in which out-of-focus images caused by multiple focal points reduce contrast. In contrast, this Fr. ICL does not create out-of-focus images and is primarily designed to reduce thickness. We emphasize that the Fresnel structure referred to here is a "refractive-type" structure that approximates a single-focus refractive surface with a stepped profile, and is distinct from a "diffractive Fresnel lens" based on interference and phase control at zone boundaries.

As shown in Fig 4, contrast reduction, which is observed in diffractive multifocal intraocular lenses, is unlikely to occur. Therefore, adopting a Fresnel structure for a hyperopic ICL was considered optically feasible for thickness reduction under the simulation framework of this study (Figs 2 and 3). Fresnel lenses have been industrially applied in various optical

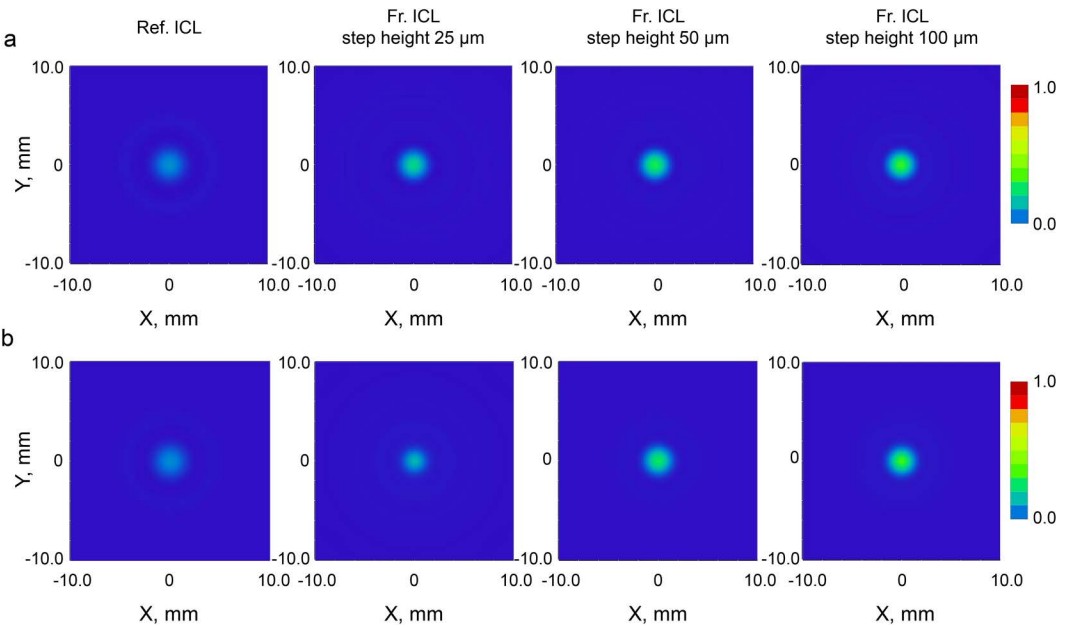

**Fig 7. Two-dimensional point spread functions (PSFs) derived from FFT calculations for the Ref. ICL and Fr. ICLs with different step heights.** (a) Monochromatic illumination at 555 nm with an artificial pupil diameter of 4.5 mm. (b) Polychromatic illumination with an artificial pupil diameter of 4.5 mm. In each row, images are shown from left to right: Ref. ICL, and Fr. ICLs with step heights of 25 µm, 50 µm, and 100 µm. The color scale ranges from 0 to 1 and represents the normalized peak intensity; lenses with lower Strehl ratios do not reach the maximum (red) intensity.

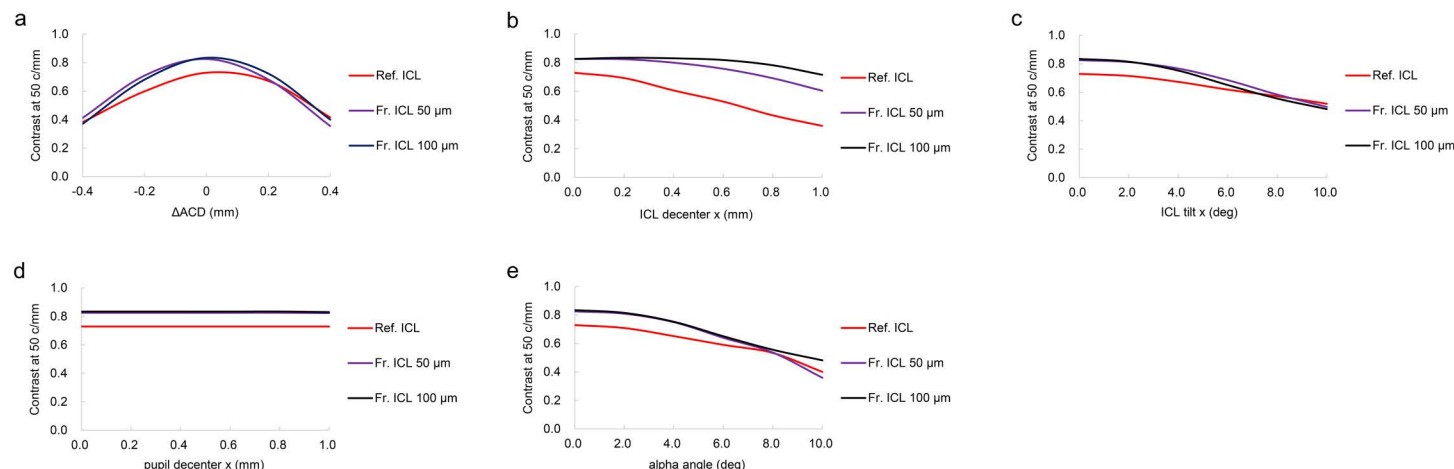

**Fig 8. Contrast at 50 c/mm for the Ref. ICL and Fr. ICLs with step heights of 50 and 100 µm under various misalignment and optical conditions.** (a) Axial displacement (ΔACD), (b) ICL decentration, (c) ICL tilt, (d) pupil decentration, and (e) alpha angle. The Ref. ICL is shown in red, and Fr. ICLs (50 µm and 100 µm) are shown in purple and black, respectively.

media, including lighthouses, and have a proven track record in multifocal IOLs, EDOF IOLs [15], and presbyopic ICLs [16], making them a technically feasible option. However, in stepped-surface structures such as the present design, wavefront discontinuities may occur at the relief edge, and some stray light may be generated near these edges. These issues

require further verification using a non-sequential illumination optical system in the future. To further investigate the effects of phase discontinuities at the Fresnel step edges, an additional analysis was performed assuming so-called tuned step height conditions, where the optical path difference between each step becomes an integer number of wavelengths at the design wavelength [24]. As a result, no significant changes were observed in the evaluated MTF and PSF metrics, and these supplementary analysis results are presented in the Supporting Information (S1 Fig in S1 File).

If the hyperopic ICL can be made thinner, it may be more suitable for eyes with a shallow anterior chamber depth, potentially supporting risk management in patient selection. Furthermore, by potentially improving usability and expanding anatomical eligibility, such a design may increase refractive correction options for high hyperopia or anisometropic hyperopia, including patient groups in which spectacle intolerance or functional complaints are common (e.g., anisometropia-related spectacle intolerance [25], amblyopia-related refractive needs [8], and digital eye strain associated with refractive burden [11]). These potential benefits should be interpreted cautiously and require further validation.

As shown in Table 2, increasing the step height leads to larger radii in the first and second zones of the Fresnel lens. When the step height decreases, the number of relief edges increases, and a slight contrast reduction was observed in areas with larger pupil diameters. Therefore, it is preferable to make the 1st zone as large as possible. In hyperopic eyes, the pupil is often small under various brightness conditions [26,27], and based on Table 2, it is expected that the number of light rays reaching the edge will not be high within the simulation range of this study. However, designing the 1st and 2nd zones according to pupil diameter is one approach, which may also offer the potential to achieve higher visual quality. Regarding the influence of holes, the Ref. hole ICL with a 3.00 mm internal artificial pupil showed lower value compared to the Ref. non-hole ICL (Fig 4). This is attributed to the increased area occupied by the hole diameter as the pupil diameter decreases, resulting in a greater impact on the MTF. This result is consistent with previous findings reported by Uozato et al. [4]. In contrast, Shiratani et al. [18] reported that the effect becomes small when the hole diameter exceeds 1.00 mm. The current hole diameter of ICLs is 0.36 mm in diameter, achieving a good balance between imaging performance and aqueous humor circulation [14]. Therefore, it is anticipated that the effect on imaging performance alone will not pose clinical issues even in hyperopic Fr. ICLs. Clinical data showed no difference in aberrations or scattering between hole ICL and non-hole ICL [28], and Shimizu et al. reported that visual function was maintained at a high level in a long-term follow-up study [3]. The hole in the ICL improves aqueous humor circulation, has a preventive effect against secondary cataracts, and eliminates the need for laser iridotomy (LI) or peripheral iridotomy (PI) [13]. The ability to suppress cataract formation with this minimal MTF reduction, while eliminating the need for LI or PI, is highly valuable, and this mechanism is also considered useful for Fr. ICLs.

The Strehl ratio results shown in Table 3 indicate that Fr. ICLs maintained performance closer to the diffraction limit compared to Ref. ICL, showing particularly high values under conditions with step heights of 50 μm or greater. This suggests that optimization of the anterior surface shape through thickness reduction led to improved wavefront quality. On the other hand, differences due to step height tended to appear more readily under large pupil conditions, indicating that the effects of residual aberrations may become manifest under conditions deviating from the design pupil.

In the monochromatic PSF shown in Fig 6, Fr. ICLs exhibited a tendency to maintain the central peak and suppress PSF spreading compared to Ref. ICL. This is consistent with the MTF and Strehl ratio results, suggesting that optimization of the anterior surface shape through thickness reduction may have contributed to light-gathering characteristics under monochromatic conditions (wavefront quality close to the diffraction limit). On the other hand, a tendency for slight broadening of the PSF tails was also observed under conditions with smaller step heights, suggesting the effect of image spreading associated with an increase in the number of steps. In the polychromatic PSF shown in Fig 7, Fr. ICLs also maintained a distribution with a dominant central peak, and the trends observed under monochromatic conditions (Fig 6) were generally preserved under white light conditions as well. Under polychromatic conditions, the effects of phase differences and aberrations that tend to manifest under monochromatic conditions are averaged due to the superposition of multiple wavelengths, allowing PSF behavior to be evaluated in a manner closer to real viewing environments. Within the scope of this study, the central peak was maintained under broadband conditions, and findings showing significant energy

redistribution to the PSF tails (strong halo-like behavior) were limited, suggesting that at least under idealized models, the image quality of Fr. ICLs is not significantly degraded even under white light conditions.

Fig 8 showed the results of evaluating sensitivity to anatomical and positioning errors that may occur in clinical practice, such as changes in anterior chamber depth (ΔACD), ICL decentration and tilt, pupil decentration, and alpha angle, using contrast at 50 c/mm. Under all conditions, Fr. ICLs (50 µm and 100 µm steps) maintained contrast equal to or greater than Ref. ICL, exhibiting relatively stable behavior against deviations from design values. Regarding ΔACD, Fr. ICLs exhibited a peak shape similar to Ref. ICL, though the contrast values at the peak were higher, showing a tendency for reduced sensitivity degradation in the vicinity of the design position. With respect to ICL decentration and tilt, Fr. ICLs showed more gradual contrast degradation, with this tendency being particularly clear in the 100 µm step design. On the other hand, with respect to pupil decentration, the effect was small for all ICLs, and the difference between Fr. ICLs and Ref. ICL was limited. With increasing alpha angle, contrast degradation was observed for all designs, while Fr. ICLs maintained higher values overall. These results suggest that thickness reduction and anterior surface shape optimization in Fr. ICLs reduce residual aberrations, and that image quality is relatively robust even against deviations from design conditions. Regarding the limitations of this study, the design values of the Ref. ICL used here were created as a pseudo-model for principle verification and may differ from those of commercially available ICLs. Because the detailed design parameters of marketed ICLs are not publicly disclosed, further validation using optical bench measurements, such as experimental MTF and stray-light evaluations, will be necessary to establish closer clinical relevance.

In addition, although the Fr. ICL in this study was designed with a central thickness of 0.40 mm, further thickness reduction may be possible from a purely optical design standpoint. In practical manufacturing and clinical use, however, the allowable lens thickness must be determined by considering manufacturing tolerances, mechanical stability, and interactions with surrounding ocular tissues, rather than optical performance alone.

Furthermore, although the present Fr. ICL is not a diffractive optical element in its operating principle, its stepped surface inherently involves geometric discontinuities that can lead to diffraction and scattering, particularly at the step edges. In this study, we combined sequential ray tracing with wave-optical analysis using FFT to evaluate image spread due to diffraction and aberrations arising from the idealized design geometry. However, this study is based solely on numerical simulation and does not include validation using prototype lenses or optical bench measurements. Therefore, scattering factors that may be problematic in real devices—such as surface microroughness, manufacturing errors at step edges, scattering within the material, and stray light from multiple reflections—cannot be adequately reproduced without non-sequential analysis, scattering models, or experimental validation. Consequently, the discussion of halos and glare in this study is limited to a qualitative description of "trends attributable to the design geometry" inferred from PSF characteristics. The results of this study should be interpreted as demonstrating the feasibility of the design and relative optical trends under idealized conditions, rather than as quantitative predictions of final optical performance in real-world environments.

## Conclusion

The Fr. hole ICL for hyperopia can reduce the central thickness of the lens while maintaining or improving optical performance under idealized simulation conditions, demonstrating the theoretical optical feasibility of this design approach. Further experimental validation and evaluation under clinically realistic conditions are required to assess its potential clinical applicability.

## Supporting information

**S1 File. Analysis of tuned Fresnel step-height conditions including S1 Fig 1–4 and S1 Table 1–2.**
(PDF)

## Acknowledgments

We thank Mr. Takehiro Yoshida for his insight into ICL lens design. Professional English-language editing was provided by Editage (a division of Cactus Communications). In addition, the authors used ChatGPT (OpenAI; GPT-5.2) and Claude Sonnet 4.5 (Anthropic) to assist with English-language editing, clarity, and wording refinement throughout the manuscript. The use of these tools was limited to language and presentation support. No independent scientific judgment, data analysis, or result generation was performed using AI. All scientific content, interpretations, and conclusions were reviewed and approved by the authors, who take full responsibility for the manuscript.

## Author contributions

**Conceptualization:** Kimiya Shimizu.

**Investigation:** Takushi Kawamorita.

**Methodology:** Takushi Kawamorita.

**Project administration:** Kimiya Shimizu, Takushi Kawamorita.

**Software:** Takushi Kawamorita.

**Supervision:** Kimiya Shimizu.

**Visualization:** Takushi Kawamorita.

**Writing – original draft:** Takushi Kawamorita.

**Writing – review & editing:** Takushi Kawamorita.

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
