## [Decision Letter · Decision Letter 0]

19 Dec 2025

PONE-D-25-48736Evaluation of optical quality in a Fresnel-structured hyperopic implantable collamer lenses using ray-tracing simulationPLOS One

Dear Dr. Kawamorita,

Thank you for submitting your manuscript to PLOS ONE. After careful consideration, we feel that it has merit but does not fully meet PLOS ONE’s publication criteria as it currently stands. Therefore, we invite you to submit a revised version of the manuscript that addresses the points raised during the review process.

We look forward to receiving your revised manuscript.

Kind regards,

Amit Kumar Goyal, PhD

Academic Editor

PLOS One

Journal Requirements:

Additional Editor Comments:

Some of the reviewers have suggested to include more literature. Authors are free to cite or not cite the mentioned references. This will not impact the decision process.

Reviewer's Responses to Questions

**Comments to the Author**

1. Is the manuscript technically sound, and do the data support the conclusions?

Reviewer #1: Partly

Reviewer #2: Partly

2. Has the statistical analysis been performed appropriately and rigorously?

Reviewer #1: N/A

Reviewer #2: No

3. Have the authors made all data underlying the findings in their manuscript fully available?

Reviewer #1: No

Reviewer #2: Yes

4. Is the manuscript presented in an intelligible fashion and written in standard English?

Reviewer #1: Yes

Reviewer #2: Yes

5. Review Comments to the Author

Reviewer #1: This is an interesting paper about reducing the thickness of a hyperopic implantable lens, but it would be worth reworking the examples to see if the image quality improves if the phase delay at the steps is forced to be an integral number of wavelengths.

L73. A concern is raised at this introductory text in the paper, because a diffractive lens is not the same as the type of “Fresnel lens” discussed here. A diffractive lens has zone boundaries that match a specific equation, with light from all the zones interfering at specific image distances that are linked by the zone boundary equation. The text here should be modified. Fresnel was involved with both lighthouse lenses that do not create an image, and diffractive effects, but not actually diffractive lenses. The wording here needs to be clearer, and Fresnel’s observation to do with diffraction was essentially that the phase repeats after every wavelength. Physical distances related to this came to be called “Fresnel zones”, but perhaps not in his lifetime?

L100 Conv. ICL. Is this conventional? Would standard be better? Conv. Is odd and not defined?

A 3mm entrance pupil is used, rather than an actual internal 3mm pupil, which would be more like an average eye. And only an additional 5 mm entrance pupil is used?

The main concern about the current paper relates to the concept described 40 years ago in this paper, and perhaps in other publications also: A "tuned" Fresnel lens. Giovanni Vannucci.15 August 1986 / Vol. 25, No. 16 / APPLIED OPTICS 2831-2834

There is also this patent from 30 years ago: United States Patent. Silberman. 5,178,636. 1993. TUNED FRESNEL LENS FOR MULTIFOCAL INTRAOCULAR APPLICATIONS INCLUDINGSMALL INCISION SURGERIES. Donn M. Silberman, Iolab Corporation, Claremont, Calif.

A recent paper that includes a discussion about diffractive lenses is the following: Design concepts for advanced-technology intraocular lenses [Invited]," Biomed. Opt. Express 16, 334 (2025). https://doi.org/10.1364/BOE.544647. There may also be other papers that discuss Fresnel lenses and pay attention to the zone boundary locations, and the phase delays at the steps. Discussions online and in the literature are often muddled about the notion of a “Fresnel lens”.

The examples in the paper have different steps in microns, and with a 50 micron step, the phase delay for a material index of 1.46 is about 10.5 waves, which means that light from one zone would destructively interfere with light from the next. A single wavelength is delayed by 0.546/(1.45-1.336), or wavelength/(index1-index2). The other physical steps have other delays.

So the fundamental question about this ICL paper is what the calculations would look like if the phase delay at each step was an integral number of wavelengths. In principle, the wavefront would then be similar to the situation with no steps at all, at the design wavelength (though the steps are high, and at an angle there would be some effect from the steps themselves). It just seems like this should be explored before this paper is published.

Reviewer #2: 1. The study relies almost exclusively on MTF analysis at a single wavelength (555 nm). While MTF is an important indicator of optical quality, it does not fully capture clinically relevant visual phenomena such as stray light, halo formation, glare, or contrast sensitivity under polychromatic conditions. The authors are strongly encouraged to extend the analysis to include additional optical quality metrics (e.g., point spread function, Strehl ratio, stray light simulation, or polychromatic MTF) to better support the claimed clinical relevance.

2. The Fresnel structure inherently introduces discontinuities that may generate diffraction effects and stray light, particularly at step edges. However, the simulations were conducted only in sequential ray-tracing mode. A non-sequential optical analysis incorporating scattering or edge diffraction effects is essential to realistically evaluate potential photic phenomena, especially given the discussion on halos and glare in hyperopic ICLs.

3. All simulations are based on a single Liou & Brennan model eye with fixed biometric parameters. Inter-individual variability in anterior chamber depth, pupil decentration, α-angle, and pupil eccentricity is not considered. A sensitivity or tolerance analysis examining the robustness of the Fresnel ICL performance under realistic anatomical variations would substantially strengthen the generalizability of the conclusions.

4. Although multiple step heights (25–100 µm) were evaluated, the rationale for selecting these specific values is not sufficiently justified from a manufacturing, biomechanical, or clinical standpoint. The manuscript would benefit from a clearer explanation of how these step heights relate to practical fabrication limits, long-term mechanical stability, and potential biological interactions within the eye.

5. The study is entirely simulation-based, and the designed conventional ICL is described as a “pseudo-model.” Without optical bench validation or experimental comparison using fabricated Fresnel structures, it is difficult to assess how well the simulation results translate to real optical performance. At minimum, the limitations of relying solely on numerical simulations should be discussed more explicitly, along with a clearer roadmap for experimental validation.

6. While the simulation results suggest improved or maintained optical performance, the manuscript occasionally implies near-clinical readiness of the Fresnel hyperopic ICL. Given the absence of experimental data, patient variability analysis, and stray-light evaluation, these claims should be tempered. The authors should more clearly distinguish between theoretical optical feasibility and clinical applicability, revising the Discussion and Conclusion accordingly.

6. PLOS authors have the option to publish the peer review history of their article (what does this mean?). If published, this will include your full peer review and any attached files.

Reviewer #1: No

Reviewer #2: No

---

## [Author Response · Author response to Decision Letter 1]

6 Feb 2026

Dear Reviewer #1:

Title: Evaluation of optical quality in a Fresnel-structured hyperopic implantable collamer lenses using ray-tracing simulation (Manuscript No. PONE-D-25-48736)

Comment #1:

1. This is an interesting paper about reducing the thickness of a hyperopic implantable lens, but it would be worth reworking the examples to see if the image quality improves if the phase delay at the steps is forced to be an integral number of wavelengths.

L73. A concern is raised at this introductory text in the paper, because a diffractive lens is not the same as the type of “Fresnel lens” discussed here. A diffractive lens has zone boundaries that match a specific equation, with light from all the zones interfering at specific image distances that are linked by the zone boundary equation. The text here should be modified. Fresnel was involved with both lighthouse lenses that do not create an image, and diffractive effects, but not actually diffractive lenses. The wording here needs to be clearer, and Fresnel’s observation to do with diffraction was essentially that the phase repeats after every wavelength. Physical distances related to this came to be called “Fresnel zones”, but perhaps not in his lifetime?

Response:

Thank you for this valuable comment. As you pointed out, reworking the design examples under the constraint of an "integer-wavelength phase wrap" condition (i.e., forcing the phase delay at the steps to be an integral multiple of 2π at the design wavelength) is important for evaluating the potential improvement in optical image quality (e.g., retinal contrast) that directly affects visual function.

In the revised manuscript, we have added a new design in which the surface profile is stepped to satisfy the 2π phase-wrapping condition at the design wavelength, and have compared its performance in terms of MTF/PSF and other metrics against the original thickness-reduced design. This has allowed us to distinguish whether performance improvements are attributable to thickness reduction or to the phase-wrapping design strategy.

We agree that the concern regarding the equivalence implied between "Fresnel lens" and "diffractive lens" is valid. In the revision, we have clarified the distinction between refractive Fresnel lenses (which use stepped refractive surfaces) and diffractive optical elements (DOE) / diffractive lenses (which utilize interference via Fresnel zones). We have also revised the introduction to avoid giving the impression that these two concepts are identical. Furthermore, we have provided a more accurate description of Fresnel's historical contributions, including the modern concept of "Fresnel zones" (phase repetition).

Comment #2:

2. L100 Conv. ICL. Is this conventional? Would standard be better? Conv. Is odd and not defined?

Response:

We appreciate the reviewer’s helpful suggestion. We agree that the abbreviation “Conv. ICL” was unclear and insufficiently defined. In the revised manuscript, we replaced it with “reference ICL (Ref. ICL)” and explicitly defined the term at its first occurrence.

Regarding terminology, we considered using “standard”; however, the comparator in this study is not a specific marketed product nor an externally defined benchmark. Instead, it serves as an internally defined reference design for within-study comparison. Therefore, we adopted the term “Ref. ICL” to clearly convey its role as a reference. We define Ref. ICL as a refractive ICL with a continuous (step-free) surface profile, i.e., without a Fresnel-type stepped structure. For clarity, this terminology has been used consistently throughout the text, figures, and tables.

Comment #3:

3. A 3mm entrance pupil is used, rather than an actual internal 3mm pupil, which would be more like an average eye. And only an additional 5 mm entrance pupil is used?

Response:

Thank you for this helpful comment. In the previous version, we described the evaluation using a “3-mm entrance pupil,” which did not clearly indicate an internal artificial pupil representing the iris plane of an average eye. In the revised manuscript, in accordance with the ISO 11979-2 framework for optical bench evaluation of intraocular lenses, we standardized the method to use an internal artificial pupil (aperture stop) placed at the iris plane of the model eye, and we evaluate optical performance under 3.0 mm and 4.5 mm pupil diameters.

Accordingly, we revised the terminology in the text and figures (entrance pupil vs. internal pupil) and recalculated the reported optical quality metrics (e.g., MTF/PSF) under these consistent conditions. Although analysis at other pupil diameters is possible, we focused on these two conditions to avoid excessive expansion of the manuscript content.

Comment #4:

4. The main concern about the current paper relates to the concept described 40 years ago in this paper, and perhaps in other publications also: A "tuned" Fresnel lens. Giovanni Vannucci.15 August 1986 / Vol. 25, No. 16 / APPLIED OPTICS 2831-2834

There is also this patent from 30 years ago: United States Patent. Silberman. 5,178,636. 1993. TUNED FRESNEL LENS FOR MULTIFOCAL INTRAOCULAR APPLICATIONS INCLUDINGSMALL INCISION SURGERIES. Donn M. Silberman, Iolab Corporation, Claremont, Calif.

A recent paper that includes a discussion about diffractive lenses is the following: Design concepts for advanced-technology intraocular lenses [Invited]," Biomed. Opt. Express 16, 334 (2025). https://doi.org/10.1364/BOE.544647. There may also be other papers that discuss Fresnel lenses and pay attention to the zone boundary locations, and the phase delays at the steps. Discussions online and in the literature are often muddled about the notion of a “Fresnel lens”.

The examples in the paper have different steps in microns, and with a 50 micron step, the phase delay for a material index of 1.46 is about 10.5 waves, which means that light from one zone would destructively interfere with light from the next. A single wavelength is delayed by 0.546 / (1.45-1.336), or wavelength / (index1 - index2). The other physical steps have other delays.

So the fundamental question about this ICL paper is what the calculations would look like if the phase delay at each step was an integral number of wavelengths. In principle, the wavefront would then be similar to the situation with no steps at all, at the design wavelength (though the steps are high, and at an angle there would be some effect from the steps themselves). It just seems like this should be explored before this paper is published.

Response:

We thank the reviewer for the valuable comments and for pointing out prior studies on tuned Fresnel lenses, including phase-synchronized Fresnel lenses reported by Vannucci and applications to multifocal intraocular lenses by Silberman. We also appreciate the reference to recent well-organized discussions on diffractive intraocular lens design.

As correctly noted by the reviewer, the so-called tuned Fresnel condition, in which the phase delay at each step corresponds to an integer multiple of the design wavelength (i.e., a 2𝜋 phase wrap), is an important consideration from a wave-optical perspective. Accordingly, in this study, we constructed an additional design model in which the step heights were adjusted so that the optical path difference between adjacent steps became an integer multiple of the design wavelength 𝜆0, and we performed a comparative evaluation.

These additional analyses were conducted under monochromatic conditions at 𝜆0 and were evaluated using FFT-based wave-optical MTF calculations. The results of these supplementary analyses are provided in the Supporting Information. Under these conditions, no substantial differences in MTF were observed between the tuned designs and the original stepped designs.

These results can be interpreted as follows. The hyperopic ICL in this study is not designed like a diffractive lens that optimizes zone boundaries to exploit interference; rather, it approximates a continuous single-focus refractive surface with a stepped profile to reduce lens thickness. Consequently, even when the phase difference between steps is matched to an "integer number of wavelengths (integral multiples of 2π)" under monochromatic conditions, the factors affecting image quality are not limited to coherent interference between adjacent zones alone. As a result, no clear change was observed in the MTF calculated by FFT.

Furthermore, we examined the phase-adjusted condition (optimized for a single wavelength) under polychromatic illumination that more closely resembles real viewing environments. Under polychromatic illumination, the integer-wavelength condition cannot be satisfied simultaneously for all wavelengths, and phase errors for each wavelength are averaged across the pupil. As a result, the design differences between phase-adjusted and non-adjusted configurations became small under most conditions. However, depending on pupil diameter and step height, differences could emerge through shifts in the best focus position, indicating that the effect of phase adjustment is condition-dependent.

Based on these considerations, the primary focus of this manuscript was placed on demonstrating the effectiveness of thickness reduction using a stepped refractive approximation. The verification of the tuned step-height condition was therefore presented as supplementary information in the Supporting Information.

Yours sincerely,

Kimiya Shimizu, MD, PhD

Department of Ophthalmology, Sanno Hospital, 8-10-16 Akasaka, Minato-ku, Tokyo, Japan, Professor

Akasaka 8-10-16, Minato-ku, Tokyo, 107-0052, Japan

Telephone: +81-3-6863-0700

Email: kimiyas@ihwg.jp

Dear Reviewer #2:

Title: Evaluation of optical quality in a Fresnel-structured hyperopic implantable collamer lenses using ray-tracing simulation (Manuscript No. PONE-D-25-48736)

Comment #1:

1. The study relies almost exclusively on MTF analysis at a single wavelength (555 nm). While MTF is an important indicator of optical quality, it does not fully capture clinically relevant visual phenomena such as stray light, halo formation, glare, or contrast sensitivity under polychromatic conditions. The authors are strongly encouraged to extend the analysis to include additional optical quality metrics (e.g., point spread function, Strehl ratio, stray light simulation, or polychromatic MTF) to better support the claimed clinical relevance.

Response:

Thank you for the important feedback from the reviewers. We agree that MTF under monochromatic conditions alone cannot adequately reflect clinically important visual phenomena such as scatter, halo, glare, or contrast characteristics under broadband conditions. Therefore, in the revised manuscript, we have added optical evaluations that are not limited to single-wavelength MTF.

Specifically, we performed FFT-based wave optics analysis using photopic luminosity function-weighted calculations for five discrete wavelengths (470, 510, 555, 610, and 650 nm) to evaluate polychromatic PSF and polychromatic MTF. The PSF was used as a practical metric to assess image spread related to scatter, halo, and glare tendencies under broadband illumination. Additionally, we evaluated the Strehl ratio as a supplementary metric. The Strehl ratio was calculated using FFT PSF in Zemax OpticStudio.

As a result, the PSF showed a clear central peak even under polychromatic conditions, with no significant energy distribution in the peripheral region (tail). This suggests that behaviors related to halos and glare do not significantly increase even under broadband illumination. These results are consistent with the MTF analysis using FFT and further support the conclusions regarding optical performance presented in this study. These additional analysis results have been added to the Results section of the main text.

Comment #2:

2. The Fresnel structure inherently introduces discontinuities that may generate diffraction effects and stray light, particularly at step edges. However, the simulations were conducted only in sequential ray-tracing mode. A non-sequential optical analysis incorporating scattering or edge diffraction effects is essential to realistically evaluate potential photic phenomena, especially given the discussion on halos and glare in hyperopic ICLs.

Response:

Thank you for this valuable and important comment. As you pointed out, Fresnel-type stepped structures inherently involve geometric discontinuities, which can lead to diffraction and scattering at the step edges. We recognize that to evaluate halos and glare in a clinically realistic manner, non-sequential ray tracing that explicitly accounts for scattering and edge diffraction, or experimental evaluation, would be necessary.

The primary objective of this study was to clarify the impact of the designed geometric structure itself on optical performance. To this end, we employed sequential ray tracing along with wave-optical analyses using FFT (MTF, PSF, Strehl ratio, and polychromatic evaluation). These analyses assess image spread due to diffraction and aberrations arising from the idealized design geometry, but do not account for scattering due to manufacturing errors, surface microroughness, imperfections at step edges, or scattering within the material.

Therefore, the discussion of halos and glare in this manuscript remains a qualitative assessment inferred from PSF trends based on the design geometry and does not quantitatively predict stray light as observed clinically. We acknowledge that more realistic evaluation, including non-sequential analysis and experimental validation, is an important future task, and we have clearly stated this limitation in the revised manuscript.

Comment #3:

3. All simulations are based on a single Liou & Brennan model eye with fixed biometric parameters. Inter-individual variability in anterior chamber depth, pupil decentration, α-angle, and pupil eccentricity is not considered. A sensitivity or tolerance analysis examining the robustness of the Fresnel ICL performance under realistic anatomical variations would substantially strengthen the generalizability of the conclusions.

Response:

Thank you for this valuable suggestion. We agree that relying on a single Liou & Brennan model eye with fixed biometric parameters can limit generalizability. To address this concern, we have performed a sensitivity/tolerance analysis to examine the robustness of the proposed Fr. ICL under realistic anatomical variations.

Specifically, while keeping the lens designs fixed (i.e., without re-optimization), we perturbed key parameters within clinically plausible ranges, including anterior chamber depth, pupil decentration/eccentricity, and α-angle–related line-of-sight offsets, and re-evaluated optical performance for both the Ref. ICL and the Fr. ICL under identical conditions. MTF was evaluated only at 50 cycles/mm as a representative spatial frequency.

The results showed that, although absolute image quality degraded with increasing anatomical variation as expected, the relative performance trends and the main conclusions of this study remained unchanged across the tested ranges. In other words, the Fresnel-inspired design did not exhibit abnormal sensitivity compared with Ref. ICL under these realistic perturbations. These new results have been added to the revised manuscript, thereby strengthening the generalizability of our conclusions.

Comment #4:

4. Although multiple step heights (25–100 µm) were evaluated, the rationale for selecting these specific values is not sufficiently justified from a manufacturing, biomechanical, or clinical standpoint. The manuscript would benefit from a clearer explanation of how these step heights relate to practical fabrication limits, long-term mechanical stability, and potential biological interactions within the eye.

Response:

Thank you for this important comment. We agree that the rationale for selecting the step height should be clearly justified from manufacturing, biomechanical, and clinical perspectives. In the revised manuscript, we have added an explanation for choosing the range of 25–100 µm.

In this study, we set this range as a realistic and conservative design space for a Fresnel ICL. From a manufacturing perspective, step heights on t

---

## [Editor Report · Decision Letter 1]

2 Mar 2026

Evaluation of optical quality in a Fresnel-structured hyperopic implantable collamer lenses using ray-tracing simulation

PONE-D-25-48736R1

Dear Dr. Kawamorita,

We’re pleased to inform you that your manuscript has been judged scientifically suitable for publication and will be formally accepted for publication once it meets all outstanding technical requirements.

Kind regards,

Amit Kumar Goyal, PhD

Academic Editor

PLOS One
---

## [Editor Report · Acceptance letter]

PONE-D-25-48736R1

PLOS One

Dear Dr. Kawamorita,

I'm pleased to inform you that your manuscript has been deemed suitable for publication in PLOS One. Congratulations! Your manuscript is now being handed over to our production team.

Kind regards,

on behalf of

Dr. Amit Kumar Goyal

Academic Editor

PLOS One